

# Seasonal and diel variations in the vertical distribution, composition, abundance and biomass of zooplankton in a deep Chilean Patagonian Fjord

Nur Garcia-Herrera[1,2], Astrid Cornils[1], Jürgen Laudien[1], Barbara Niehoff[1], Juan Höfer[3,4,5], Günter Försterra[3], Humberto E. González[4,6] and Claudio Richter[1,2]

[1] Alfred Wegener Institute, Helmholtz Centre for Polar and Marine Research, Bremerhaven, Germany
[2] University of Bremen, Bremen, Germany
[3] Escuela de Ciencias del Mar, Pontificia Universidad Católica de Valparaíso, Valparaíso, Chile
[4] Research Center: Dynamics of High Latitude Marine Ecosystems (IDEAL), Punta Arenas, Chile
[5] Fundación San Ignacio de Huinay, Huinay, Chile
[6] Institute of Marine and Limnological Sciences, Universidad Austral de Chile, Valdivia, Chile

Corresponding author
Nur Garcia-Herrera,
nur.garcia.herrera@awi.de

## ABSTRACT

Comau Fjord is a stratified Chilean Patagonian Fjord characterized by a shallow brackish surface layer and a >400 m layer of aragonite-depleted subsurface waters. Despite the energetic burden of low aragonite saturation levels to calcification, Comau Fjord harbours dense populations of cold-water corals (CWC). While this paradox has been attributed to a rich supply of zooplankton, supporting abundance and biomass data are so far lacking. In this study, we investigated the seasonal and diel changes of the zooplankton community over the entire water column. We used a Nansen net (100 μm mesh) to take stratified vertical hauls between the surface and the bottom (0-50-100-200-300-400-450 m). Samples were scanned with a ZooScan, and abundance, biovolume and biomass were determined for 41 taxa identified on the web-based platform EcoTaxa 2.0. Zooplankton biomass was the highest in summer (209 g dry mass m$^{-2}$) and the lowest in winter (61 g dry mass m$^{-2}$). Abundance, however, peaked in spring, suggesting a close correspondence between reproduction and phytoplankton spring blooms (Chl *a* max. 50.86 mg m$^{-3}$, 3 m depth). Overall, copepods were the most important group of the total zooplankton community, both in abundance (64–81%) and biovolume (20–70%) followed by mysids and chaetognaths (in terms of biovolume and biomass), and nauplii and Appendicularia (in terms of abundance). Throughout the year, diel changes in the vertical distribution of biomass were found with a daytime maximum in the 100–200 m depth layer and a nighttime maximum in surface waters (0–50 m), associated with the diel vertical migration of the calanoid copepod family Metridinidae. Diel differences in integrated zooplankton abundance, biovolume and biomass were probably due to a high zooplankton patchiness driven by biological processes (*e.g.,* diel vertical migration or predation avoidance), and oceanographic processes (estuarine circulation, tidal mixing or water column stratification). Those factors are considered to be the main drivers of the zooplankton vertical distribution in Comau Fjord.

## INTRODUCTION

Coastal marine ecosystems are among the most productive on earth (*Mann & Lazier, 1991*). They provide substantial economic and ecological services, such as high biological production, nutrient cycling or shoreline stability and erosion control (*Escribano, Fernández & Aranís 2003*; *Liu et al., 2010*; *Pan et al., 2013*; *Barbier, 2017*). The Chilean Fjord region extends over large and complex hydrographic and geomorphologic conditions, particularly rich in productivity and biodiversity (*Försterra, Häussermann & Laudien, 2017*; *Häussermann, Försterra & Laudien, 2021*), but also endangered by human exploitation, which has been increasing significantly—*e.g.*, salmon aquaculture—over the last two decades (*Iriarte, González & Nahuelhual, 2010*; *Buschmann, Niklitschek & Pereda, 2021*; *Navedo & Vargas-Chacoff, 2021*).

The hydrography of the Fjords and channels of Patagonia can be considered a transitional estuarine-marine system where a surface layer of silicate-rich terrestrial freshwater meets nitrate- and phosphate-rich marine waters. These Fjords receive freshwater from rivers, surface runoff and groundwater flow due to the high rainfall and glacier melting (*Pantoja, Iriarte & Daneri, 2011*). The upper brackish layer within the top 10 m water depth is usually poor in nitrate and phosphate but rich in silicate and organic matter from terrestrial inputs (*Sánchez, González & Iriarte, 2011*). Below the halocline, a water mass with higher salinity (>31), named Modified Subantarctic Water (MSAAW), flowing landward from the adjacent oceanic area provides the Fjords with macronutrients (nitrate and phosphate). Both water masses, surface-freshwater and MSAAW, generate a two-layer structure in the water column with sharp vertical and more gradual horizontal salinity gradients (*Sievers & Silva, 2008*; *Pérez-Santos et al., 2014*; *Meerhoff et al., 2019*). The summer stratification creates a barrier which may hinder the exchange of nutrients within the water column, altering the functioning of the pelagic food web and productivity patterns (*Silva, Calvete & Sievers, 1997*; *González et al., 2011*). During austral spring, the Comau Fjord receives an intense riverine input of fresh water, rich in silicic acid used by bloom-generating diatoms and, thus, leading to high primary production (*González et al., 2010*). In austral summer, the high concentration of phytoplankton promotes an increase in the abundance and biomass of zooplankton as secondary producers (*Antezana, 1999*; *González et al., 2010*). In Fjord systems, seasonal patterns are modulated by other oceanographic processes, such as estuarine circulation (*Palma & Silva, 2004*), tidal regimes and lateral advection (*Castro et al., 2011*) or water column stratification (*Sánchez, González & Iriarte, 2011*) influencing the zooplankton biomass and community structure on shorter time scales.

Zooplankton plays an essential role in the functioning of marine ecosystems and in the oceanic carbon cycle. It includes a wide variety of organisms and displays extreme variability in terms of community composition and vertical, seasonal and geographical distribution (*Palma & Kaiser, 1993*). Many taxa are known to perform diel vertical migrations (DVM),

most likely to evade predators (*Stich & Lampert, 1981*; *Iwasa, 1982*). According to the predator-evasion hypothesis, migrating zooplankton resides in deep waters during daytime hours where the probability of being perceived by visually orientated predators is lower than if they remained in better illuminated shallow waters, and at night, in the refuge of darkness, they migrate upwards to feed. However, DVM is not performed by all organisms in a zooplankton community or even not by all individuals of one species. For example, while late copepodites and adults of the copepod genus *Metridia* migrate, a large fraction of the young developmental stages remains in surface waters, saving the energy of performing the DVM, suggesting a lower probability of being perceived and consumed by visual predators (*Hays, 1995*). Both, migrant and non-migrant species are important elements of the biological carbon pump *via* the production of sinking fecal pellets that transport carbon from surface waters to the seafloor (*Urrère & Knauer, 1981*; *Fowler & Knauer, 1986*; *Emerson & Roff, 1987*). Zooplankton also provides a trophic link between primary production and higher consumers such as fish, birds and mammals, but also invertebrate predators, such as corals (*Nemoto, 1970*; *Gili et al., 2006*; *Höfer et al., 2018*).

Cold-water corals (CWC) rely on zooplankton as their principal food source to maintain their physiological processes, such as respiratory metabolism and growth (*Carlier et al., 2009*; *Mayr et al., 2011*; *Naumann et al., 2011*). Therefore, their diet depends on the zooplankton seasonal, diel and vertical distribution. In Comau Fjord, azooxanthellate scleractinian CWC are wide-spread even in deep aragonite-undersaturated waters (*Häussermann & Förstera, 2007*; *Fillinger & Richter, 2013a*; *Jantzen et al., 2013a*). In the latter, the dissolution of exposed parts of the skeleton, enhanced bioerosion, and reduced CWC growth and survival have been observed (*McCulloch et al., 2012*; *Maier et al., 2016*). The calcification of the CWC skeleton is energetically costly, and thus food requirements in these adverse environments are higher compared to aragonite-saturated waters (*Maier et al., 2016*). According to *Fillinger & Richter (2013b)*, in Comau Fjord, the CWC *Desmophyllum dianthus* (Esper, 1794) thrives but coral densities decrease below 280 m despite available substrate, suggesting that lower oxygen and pH concentrations, combined with a shortage of zooplankton could be limiting coral growth. However, up to now little is known about the zooplankton of Comau Fjord in the northern Patagonian region and its role in sustaining the CWC communities living in the Fjord. Most studies have been carried out in the central-southern part of Patagonia (from Penas Gulf to Cape Horn, S46.50°–S55.55°), while studies performed in the northern area (from Puerto Montt to San Rafael Lagoon, S41.20°–S46.40°) mainly focused on selected microzooplankton taxa in the upper water column, on bulk measurements of zooplankton biomass, or on the carbon flow through the pelagic food web (*e.g.*, *Palma, 2008 Villenas, Soto & Palma, 2009*; *González et al., 2010*; *González et al., 2011*; *Palma et al., 2011*; *Sánchez, González & Iriarte, 2011*). Other studies addressed the physical oceanographic processes and their effects on zooplankton distribution (*e.g.*, *Marín & Delgado, 2009*; *Castro et al., 2011*), and their relationship with zoo- and ichthyoplankton growth and feeding (*Landaeta et al., 2015a*; *Landaeta et al., 2015b*). Recent studies have investigated zooplankton migration patterns by acoustic backscatter and vertical velocity profiles (*Valle-Levinson et al., 2014*; *Díaz-Astudillo, Cáceres & Landaeta, 2017*; *Pérez-Santos et al., 2018*). The information on
zooplankton diversity and migration patterns is, however, still very fragmentary and the linkage between the abundance of CWC and zooplankton supply in Comau Fjord remains unknown.

In this study, we aim to describe the diel, vertical, and seasonal distribution of mesozooplankton groups of Comau Fjord, with a focus on the dominating taxa, particularly those that migrate and are more likely to aggregate. Samples were collected at day and night hours with vertical net hauls through the whole water column in spring, summer, autumn and winter. They were processed with a high-resolution image analysis system (ZooScan, *Gorsky et al., 2010*). The zooplankton taxa were identified using EcoTaxa 2.0 (*Picheral, Colin & Irisson, 2017*), allowing to assess the influence of seasonal environmental changes on zooplankton dynamics, and the food naturally available to CWC.

## MATERIALS & METHODS

Field work was carried out in Comau Fjord, Northern Patagonia, Chile (Fig. 1). Zooplankton was sampled four times at a fixed station (42°14.95S, 72°28.83W) in central Comau Fjord: in austral spring (28th September 2016; three days before spring tide-new moon), summer (17th January 2017; two days after spring tide-full moon), autumn (7th May 2017; three days before spring tide-full moon), and winter (29th July 2017; one day before neap tide), during both day (noon) and night (midnight). Samples were collected with a 70 cm-diameter Nansen closing net (mesh size: 100 $\mu$m) equipped with a non-filtering cod end. Vertical hauls were carried out at 0.45 m s$^{-1}$ to sample the depth strata 0-50-100-200-300-400-450 m. Immediately after the collection, the samples were sieved through a 50 $\mu$m mesh and preserved in 4% borax-buffered formaldehyde for laboratory analyses. After every zooplankton haul, a CTD multi-probe (SBE 19plusV2 Profiler - with RS 232 Interface, Sea-Bird Electronics Inc.) was deployed from the surface to the bottom, measuring conductivity, temperature, oxygen, pH and chlorophyll *a* (Chl *a*)- fluorescence.

In the laboratory, fixed zooplankton samples were washed with fresh water and prepared for analysis with a ZooScan digital imaging system (*Grosjean et al., 2004*; *Gorsky et al., 2010*). ZooScan (CNRS patent, http://www.hydroptic.com) provides a quick and reliable method for the analysis of preserved plankton samples, storing digitized images for later examination, reprocessing and dissemination. Concentrated samples were subsampled with a Folsom plankton splitter to avoid images being cluttered with more than approximately 1,000–1,500 individuals. Up to six binary splitting steps were carried out (corresponding to a minimum 1/64th fraction of the original sample). Routinely, the two final splits were scanned with ZooScan yielding images of 2,400-dpi resolution (14,200 × 22,700 pixels). The hinged base of the ZooScan allowed the recovery of the complete undamaged subsample, which was later stored in 70% ethanol for archiving. Most overlapping individuals on the scanning surface were manually separated to ensure an even distribution before scanning. Image analysis was performed with the software ZooProcess (*Gorsky et al., 2010*), a plug-in for the image processing and analysis software ImageJ (*Schneider, Rasband & Eliceiri, 2012*). The processing involved (1) the automatic subtraction of background noise, (2) the

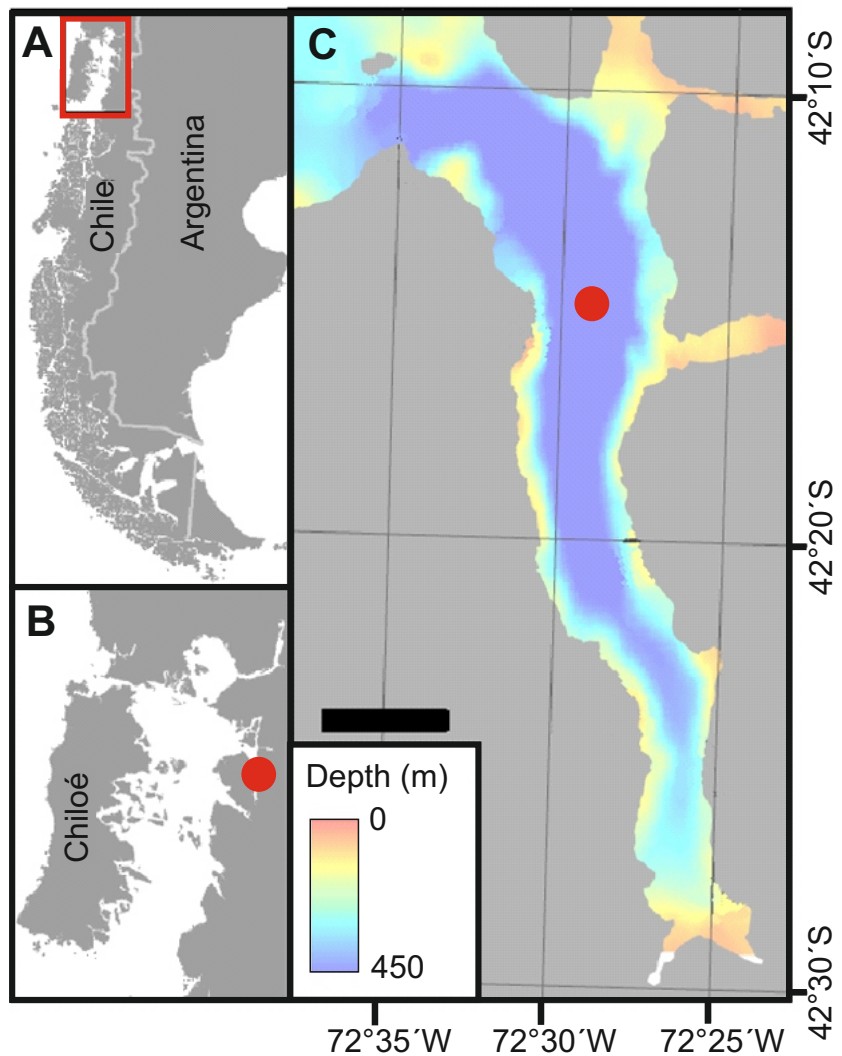

**Figure 1  Study site.** (A) Overview of Chilean Patagonia. Red square denotes area in B. (B) Inner Sea off Chiloé island, where Comau Fjord (red dot) is located. (C) Comau Fjord with the bathymetry and location of the station where zooplankton samples were taken (red dot). Adapted from *Fillinger & Richter (2013a)*.

automatic thresholding and detection of objects, and (3) the automated storage of detected objects in separate images ("vignettes"). Below 300 µm, organisms were often too blurred to be identified. Thus, the ZooScan detection limit was set at the standard of 300 µm so that detected zooplankton sizes ranged from 0.3 to 59 mm. The automatic processing of the scans was successful in 75–80% of the cases where vignettes with one individual were obtained. However, despite the manual separation, some individuals overlapped, resulting in vignettes with two or more objects. These objects on the pictures were manually separated using the "separation with mask" tool of the ZooProcess software. Separated vignettes were stored, while the original vignette, containing multiple objects was eliminated from the database to avoid duplicate counts. In some cases, the separation of individuals was not

possible as cutting the vignette would mean losing information about the morphology of the organisms (*i.e.,* cutting overlapping urosomes from two different copepods or small copepods embedded in cnidarians). Overall, the contribution of vignettes with multiple objects that could not be separated was always <10% of the total amount of vignettes.

Vignettes were subjected to the semi-automated taxonomic classification in EcoTaxa 2.0 (*Picheral, Colin & Irisson, 2017*). This web-based machine learning application uses training sets of expert-identified taxa and random forest classification to automatically identify and sort the objects. Although EcoTaxa contains more than 160 million objects on its server, no ZooScan training set was available for Patagonian waters. Therefore, manual identification of individuals on a subset of the images was first necessary to train an initial model, which was later used by the system to classify the scanned organisms. The initial learning set with Patagonian organisms improved progressively its prediction by sorting more objects into the given categories. This produced the final learning set for the classification of the entire image data set. At the end, all classified objects were individually validated to assure a correct classification. The organisms were classified to the lowest possible taxonomic level; for most copepods this was the family level. However, small calanoid copepods (<1.5 mm) were not distinguishable on family level and were thus comprised as one category: "Calanoida (<1.5 mm)" including five groups (copepodites (all calanoid taxa, <1 mm), Clausocalanidae, Microcalanidae, small Calanidae (*Neocalanus* spp.) and Paracalanidae). Developmental stages were included in the corresponding taxon as long as they were clearly identifiable. Only for calanoid copepods, the classification of some developmental stages was not clear and therefore, they were included into the category "copepodites", which included developmental stages of different calanoid copepod taxa. The category Cnidaria was constituted by organisms from the class Hydrozoa (mostly medusa and Siphonophorae). Another category contained all images that were out of focus ("bad focus") and likely comprised individuals from all copepod taxa, in total 6,766 vignettes. From the total of 83,516 vignettes, 23,227 could not be assigned to zooplankton taxa, but were labelled as "detritus", "feces", "fiber", "leg" "bubble" and "other" and were not considered in our analyses. ZooProcess provides information about the length and width of each object, allowing the calculation of its volume as a proxy for its biomass (*Gorsky et al., 2010*). The program automatically fits an ellipse around the object, from which the major and minor axis and volume (V) is computed:

$$V(mm^3) = \frac{4}{3} \times \pi \times \frac{major\ axis\ (mm)}{2} \times \left(\frac{minor\ axis\ (mm)}{2}\right)^2$$

Biovolume (BV) was then calculated as the sum of the volumes of all objects ($\Sigma$V) divided by the fraction of the sample (*e.g.,* F = 1/64) and by the volume filtered by the Nansen net ($V_N$):

$$BV\left(mm^3/m^3\right) = \frac{\left[\frac{\sum V(mm^3)}{F}\right]}{V_N(m^3)}$$

$V_N$ was calculated as:

$$V_N\left(m^3\right) = \left[\pi \times \left[\frac{net\ diameter\ (m)}{2}\right]^2\right] \times depth\ interval\ (m) \times filtration\ efficiency$$

**Table 1 Regression coefficients between individual dry mass and body area to estimate biomass [DM ($\mu$g) = ($a\,A^b$)] for different groups given by *Lehette & Hernández-León (2009)*.** Area provides the size range for each category observed in this study.

| Organism | *a* | *b* | Area (mm²) |
| --- | --- | --- | --- |
| Actinopterygii (eggs and larvae) | 43.38 | 1.54 | 0.079–1.198 |
| Appendicularia | 43.38 | 1.54 | 0.056–6.071 |
| Ascidiacea (larvae) | 43.38 | 1.54 | 0.072–1.652 |
| Amphipoda | 43.38 | 1.54 | 0.103–59.854 |
| Brachiopoda (larvae) | 43.38 | 1.54 | 0.193–0.366 |
| Bivalvia (larvae) | 43.38 | 1.54 | 0.071–2.040 |
| Bryozoa (larvae) | 43.38 | 1.54 | 0.067–0.240 |
| Chaetognatha | 23.45 | 1.19 | 0.068–15.935 |
| Cirripedia (larvae and cypris) | 43.38 | 1.54 | 0.071–0.286 |
| Cladocera | 43.38 | 1.54 | 0.072–0.455 |
| Cnidaria | 4.03 | 1.24 | 0.051–95.743 |
| Copepoda | 43.97 | 1.52 | 0.068–9.177 |
| Decapoda (zoea) | 43.38 | 1.54 | 0.072–6.733 |
| Echinodermata | 43.38 | 1.54 | 0.070–0.757 |
| Euphausiacea | 43.38 | 1.54 | 0.145–461.813 |
| Eggs | 43.38 | 1.54 | 0.070–1.952 |
| Gastropoda (larvae) | 43.38 | 1.54 | 0.071–2.266 |
| Isopoda | 43.38 | 1.54 | 0.073–0.930 |
| Mysidacea | 43.38 | 1.54 | 0.126–43.504 |
| Nemertea (pilidium) | 43.38 | 1.54 | 0.082–0.777 |
| Ostracoda | 43.38 | 1.54 | 0.066–1.270 |
| Platyhelminthes (larvae) | 43.38 | 1.54 | 0.075–0.162 |
| Polychaeta (larvae) | 43.38 | 1.54 | 0.068–7.535 |

where filtration efficiency was assumed as the theoretical 100% efficiency (value = 1). Flowmeter readings were not used because the speed of the net haul was below the measuring range of the mechanical flowmeter.

For the estimation of biomass, a regression between the dry mass (DM) of a specimen and its body area [DM ($\mu$g) = ($a\,A^b$)] was used (*Hernández-León & Montero, 2006*; *Lehette & Hernández-León, 2009*), where $A$ is the area (mm²) of each scanned individual. The regression required different conversion factors depending on the organism, as for instance, gelatinous zooplankton with high water content may not be compared to crustaceans or echinoderms (Table 1). Such coefficients have been successfully published in previous studies for mid-latitude shelf areas (*Marcolin, Gaeta & Lopes, 2015*) or the Chilean upwelling region (*Tutasi & Escribano, 2020*).

The biomass (B) of each taxon was then calculated as the sum of the individual dry masses of the respective taxon ($\Sigma$DM) divided by the fraction of the sample (*e.g.*, F = 1/64) and by the volume filtered by the Nansen net ($V_N$):

$$B\left(mg\ dry\ mass/m^3\right) = \frac{\left[\frac{\Sigma DM\,(mg)}{F}\right]}{V_N\,(m^3)}.$$

Biovolume and biomass were calculated to obtain the sum of the values of all individuals for a given taxon. In multiple vignettes, the automatic calculation of biovolume and biomass was not possible because of overlapping specimens from different taxa. Then, the organisms were counted manually and biovolume and biomass were estimated by multiplying the mean volume or DM of the given taxon from all automatic calculations by the extra number of multiple vignettes. Groups with large size variability (*e.g.*, chaetognaths, cnidarians or Euchaetidae) were divided into two categories, small (0.003–4.242 mm$^3$) and large (4.243–90.083 mm$^3$) in order to get a better biovolume/biomass assessment.

The integrated values of abundance, biovolume and biomass, were calculated down to 400 m water depth for all seasons, taking out the last 50 m from summer and winter, to make it comparable to the spring and autumn seasons, where samples were collected down to 400 m.

The relationships among physicochemical variables and each taxon's abundance were analyzed using a redundancy analysis (RDA). RDA is a constrained ordination procedure which allows the assessment of how much of the variation of one set of response variables (*i.e.,* zooplankton abundances) is explained with another set of variables (*i.e.,* physicochemical variables). The RDA is a multiresponse analysis which summarizes the linear relationships among dependent and independent variables into a matrix followed by a principal component analysis (PCA). Mean values of temperature, salinity, oxygen, and Chl *a* (log transformed) for the entire water column were used as explanatory variables. RDA was performed in R (*R Core Team, 2018*) by using the *rda* function of the *vegan* package (*Oksanen et al., 2019*). The problems caused by non-normal distributions in testing the significance of RDA results were solved by a permutation test (10,000 iterations) (*Borcard, Gillet & Legendre, 2011*) using the *anova.cca* function from the *vegan* package. All abundance data were logarithmically transformed before analysis.

The centroid depths (CD) of the zooplankton groups for each sampling event were calculated as: $CD = \sum (p_k \times z_k) / \sum p_k$, where $p_k$ is the number of organisms in the stratum $k$, and $z_k$ is the mean depth of the stratum $k$. CD were calculated for abundance and biovolume–not biomass–to better represent the gelatinous zooplankton groups (*e.g.*, Cnidaria). Due to the lack of replication, these values were compared using a contingency table by means of a Chi-square test, in order to test the significance of night-day changes in CD. The Chi-square test compares the critical values to assess their significance according to their degrees of freedom $(df) = (r - 1)(c - 1)$, where $r$ is the number of rows and $c$ the number of columns in the contingency table. After showing significant day-night differences in CD, a dissimilarity analysis (function *simper* from *vegan* package) performed pairwise comparisons of zooplankton taxa and estimated the average contribution of each taxon to the average overall Bray-Curtis dissimilarity. We listed the zooplankton taxa which cumulatively contributed at least to 70% of the night-day differences observed.

## RESULTS

The physicochemical parameters measured throughout the water column of Comau Fjord showed a stronger seasonal variability in surface waters (0–50 m) than in deep waters

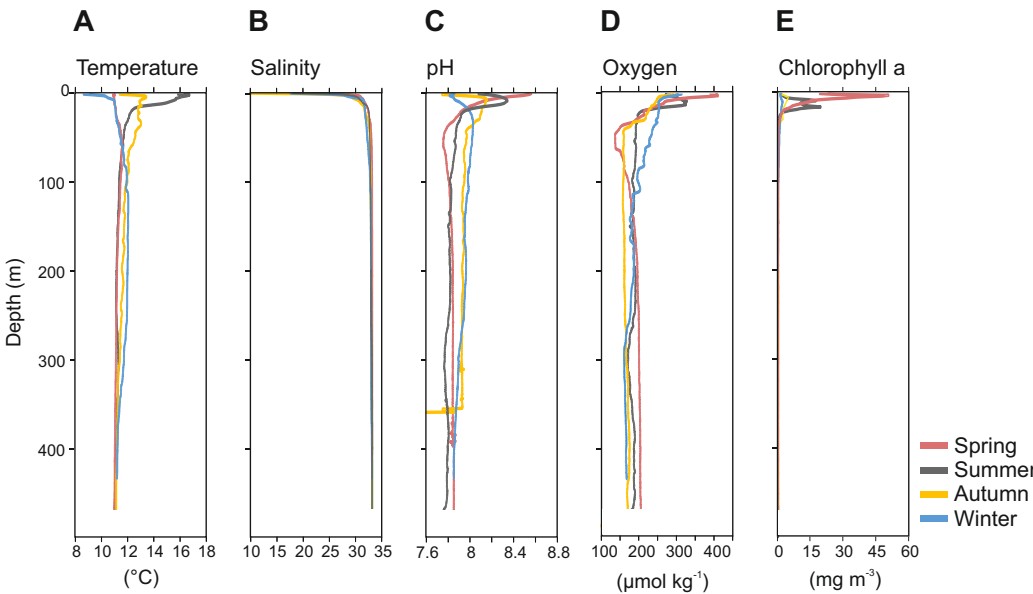

**Figure 2** **Vertical profiles of physico-chemical parameters of Comau Fjord.** (A) Temperature, (B) salinity, (C) pH, (D) oxygen and (E) Chlorophyll–*a*.

(50–450 m) (Fig. 2). The temperature profile indicated summer stratification down to 17 m water depth, followed by surface cooling, breakdown of the thermocline in autumn, and reverse temperature gradients in winter and spring. Accordingly, the surface temperature values were the lowest in winter and spring (8.6–11 °C), and the highest in summer (16.7 °C), getting cooler again in autumn (12 °C). In deeper waters, temperatures were more stable with an average value of 11.4 ± 0.2 °C (mean ± SD; Fig. 2A). Salinity was between 10-30 in the upper 20 m and 32.9 ± 0.4 below 20 m (Fig. 2B). The pH ranged between 8.5 and 7.7 in the upper 50 m and was 7.9 ± 0.1 in deeper waters in all seasons, except for autumn, where we interpret the sudden drop of pH values as an instrument malfunction (Fig. 2C). Oxygen concentration showed the largest variations in the upper 50 m during the spring season (137.2–410.5 $\mu$mol kg$^{-1}$), while below 100 m depth it was on average 180 ± 9.3 $\mu$mol kg$^{-1}$ (Fig. 2D). The chlorophyll *a* (Chl *a*) concentration peaked in early spring (50.86 mg Chl *a* m$^{-3}$ at 3 m depth), followed by a decrease towards the end of the summer and low values through autumn and mid-winter (2.5–5 mg Chl *a* m$^{-3}$ at 5–10 m depth) (Fig. 2E). Below 25 m, the concentration of Chl *a* was <1.8 ± 0.5 mg m$^{-3}$ throughout the year (Fig. 2E).

The zooplankton community exhibited large seasonal and diel differences. Abundance, integrated over the upper 400 m of the water column (individuals m$^{-2}$) showed the highest values in spring and the lowest during autumn for both day and night (Fig. 3A). The integrated biovolume (cm$^3$ m$^{-2}$) and biomass (g dry mass m$^{-2}$) showed a different pattern with the highest values in summer and the lowest in autumn and winter (Figs. 3B, 3C). Diel differences in integrated abundances generally showed higher values during the day than at night, except for autumn (Fig. 3A). Diel differences in integrated biovolume and

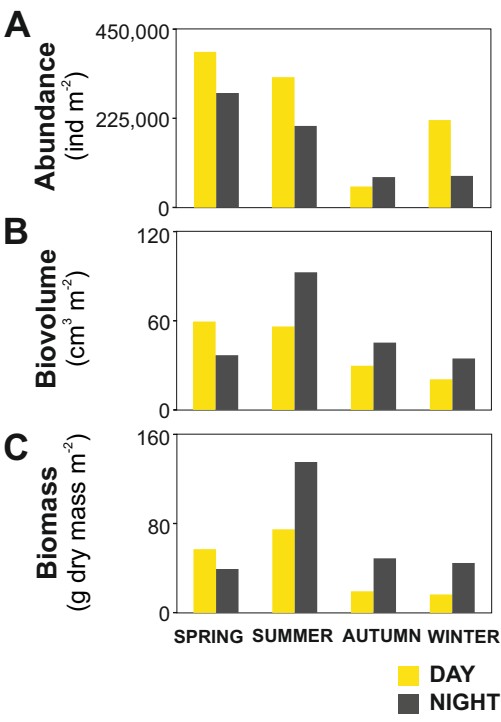

**Figure 3** **Zooplankton seasonal and diel distribution.** Seasonal distribution of integrated (A) abundance, (B) biovolume and (C) biomass of the zooplankton community during day and night.

biomass were surprisingly large, with generally higher values during the night, except for spring (Figs. 3B, 3C), indicating that during daytime, particularly in spring, zooplankton was more numerous but smaller in size.

Overall, the centroid depth dissimilarity analysis showed that in spring, fewer taxa contributed to the significant day-night differences (Chi-square Pr < 2.2 e$^{-16}$, $p < 0.001$) both in abundance and biovolume, while in summer and autumn the number of contributing zooplankton groups increased (Fig. 4). The zooplankton vertical distribution showed that abundance exhibited the highest values during day and night in the 0–50 m layer at all seasons, with the exception of autumn during day time, where abundances showed low values throughout the water column (Fig. 5A). The lowest abundances were found in >300 m depth in spring, summer and winter, and in 50-100 m depth in autumn during day and night time (Fig. 5A). The highest biovolume and biomass daytime values were observed in 0–50 m and 100–200 m water depth in spring and summer, and in 100–300 m in autumn. However, in winter, daytime biovolume and biomass were similarly low throughout the entire water column (Figs. 5B, 5C). At night, the highest biovolume and biomass values were found at the surface (0–50 m) in all seasons. The lowest biovolume and biomass values were found in >300 m depth in spring and summer, in 50–100 m and 200–300 m depth in autumn and in >50 m depth in winter (Figs. 5B, 5C).

The taxonomic composition greatly differed when considering biovolume and abundance (Figs. 6, 7A; biovolume—not biomass—was used to better represent the

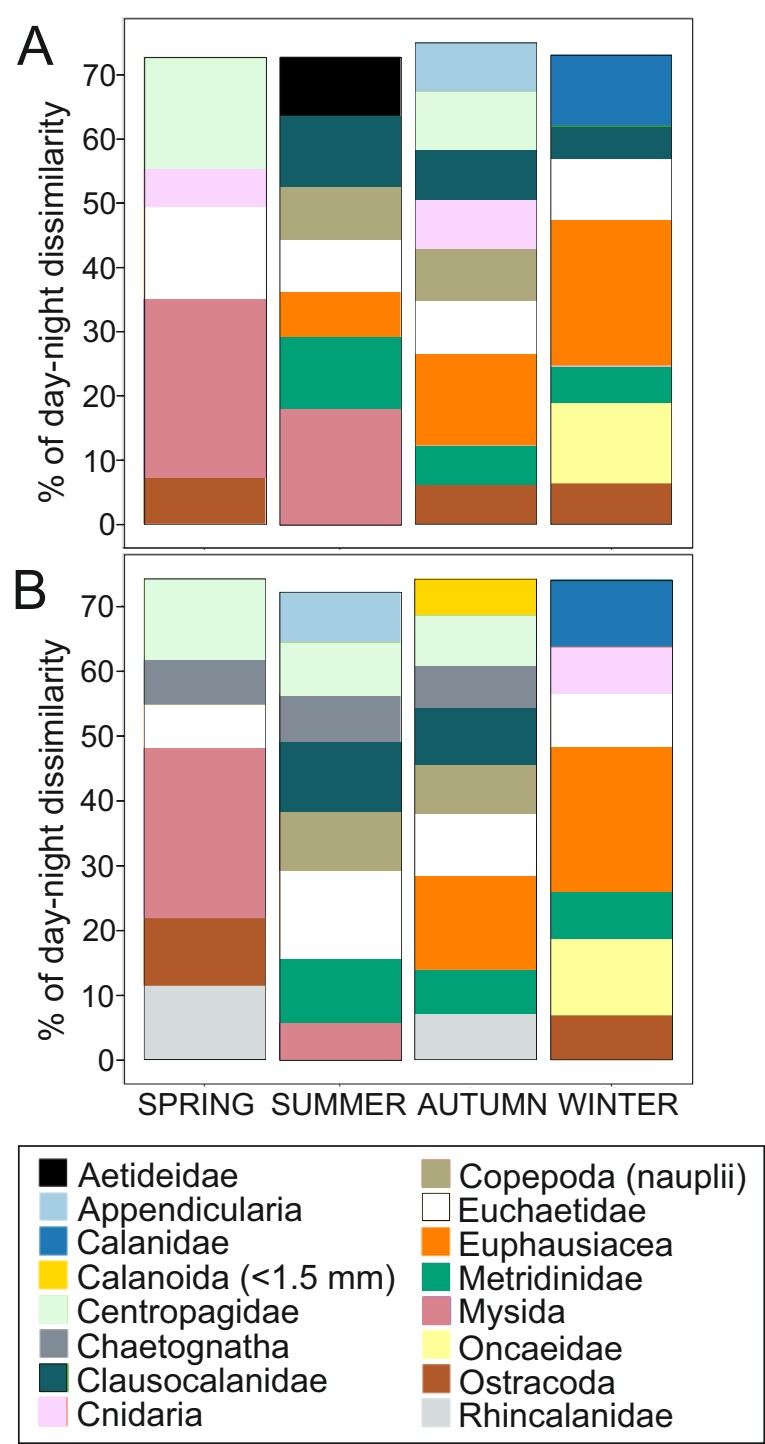

**Figure 4 Dissimilarity analysis of day-night differences in centroid depth (CD).** Zooplankton taxa which contribute cumulatively to at least 70% of the day-night differences in (A) abundance CD and (B) biovolume CD.

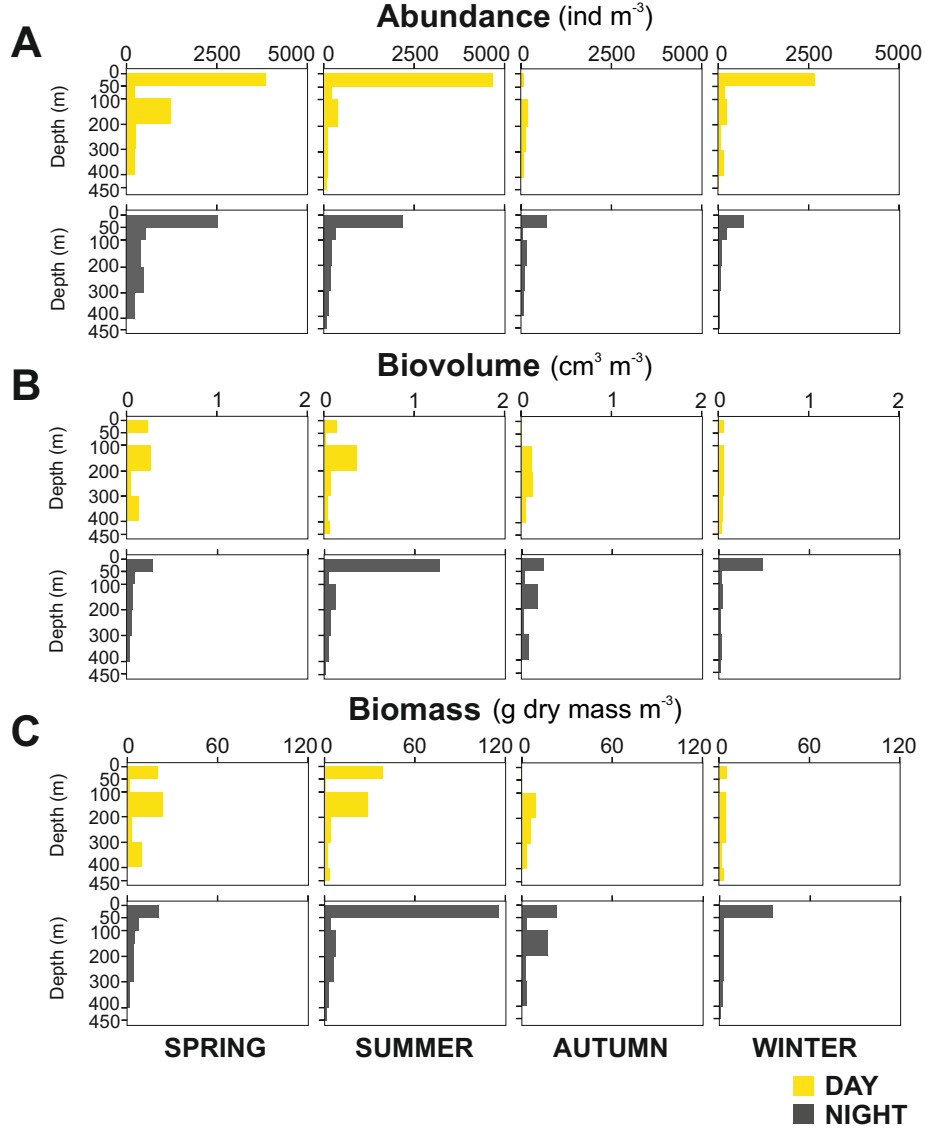

**Figure 5   Zooplankton vertical distribution.** Seasonal, diel and vertical distribution of (A) abundance, (B) biovolume and (C) biomass of the zooplankton community during day and night.

gelatinous taxa). Copepoda were generally the dominant group, constituting 20–70% of the total biovolume, and 64–81% of the total abundance. Within the copepod community, individuals smaller than 1.5 mm included (a) copepodites and adults of small calanoid genera, such as Clausocalanidae, Microcalanidae, Neocalanidae and Paracalanidae; (b) cyclopoids of the genera *Oithona* and *Oncaea*; (c) harpacticoids; and (d) nauplii. These small copepods accounted for 58–86% of the total abundance within the copepod community. Overall, 14 out of 41 taxa contributed 45–98% of the total biovolume and 45–86% of the total zooplankton abundance (Fig. 6). The other 27 taxa were constituted by copepods (Acartiidae, Aetideidae, Calanoida (non-identifiable),

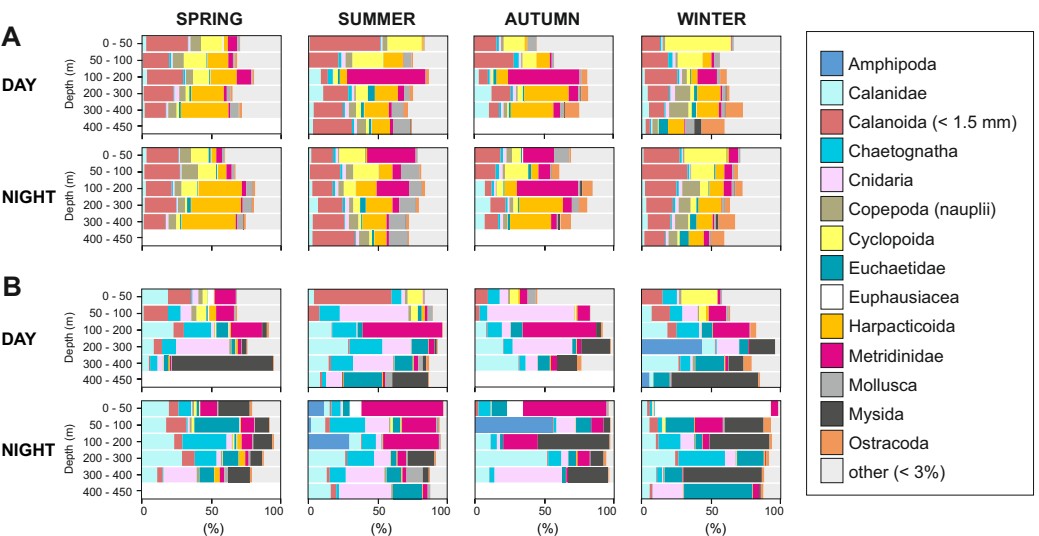

**Figure 6 Group dominance of the mesozooplankton taxa.** Relative (A) abundance and (B) biovolume of major zooplankton groups. Taxa comprising less than 3% of the total zooplankton community (27 taxa) were pooled together as "other".

Candaciidae, Centropagidae, Copepoda (non-identifiable), Eucalanidae, Heterorhabdidae, Lucicutiidae, Oncaeidae, Pontellidae, Rhincalanidae), Actinopterygii (eggs and larvae), Appendicularia, Ascidiacea (larvae), Brachiopoda (larvae), Bryozoa (larvae), Cirripedia (nauplii and cypris), Cladocera, Decapoda (larvae), Echinodermata (larvae), eggs, Isopoda, Nemertea (larvae), Platyhelminthes (larvae), Polychaeta (larvae) and non-identifiable organisms.

The contribution of groups other than copepods to the total zooplankton community differed regarding biovolume and abundance. For biovolume, Cnidaria (2–23%) and Mysida (1–21%) constituted a large part of the zooplankton community, followed by Chaetognatha (3–15%). Metanauplii and calyptopis stages of Euphausiacea were found mostly in spring and summer, accounting for 0–7% of the total biovolume. A single adult specimen found during winter in 0–50 m water depth at night raised the total biovolume of this taxon to 65%. Regarding abundance, copepod nauplii (2–9%, with their maximum in spring and minimum in autumn) and Appendicularia (0.4–6%) were the second and third most abundant groups after Copepoda, respectively. The fourth most abundant group differed among seasons: Echinodermata larvae (5% in spring), Mollusca larvae (2–5% in summer, 3–9% in autumn), Ostracoda (4–5% in autumn) and Bryozoa larvae (2–10% in winter). Across all samples and seasons, taxa that represented more than 5% of total biovolume were Cnidaria (13%), Calanidae (12.9%), Mysida (12.7%), Metridinidae (12.6%), Chaetognatha (9.6%) and Euchaetidae (8.7%). The most abundant groups with more than 5% of the total abundance were Harpacticoida (14.0%), Cyclopoida (9.1%) and Metridinidae (8.4%) (Figs. 6, 7A).

The integrated abundance of small and large copepods and chaetognaths showed the highest abundance in spring and generally low values in autumn and winter (Table 2).

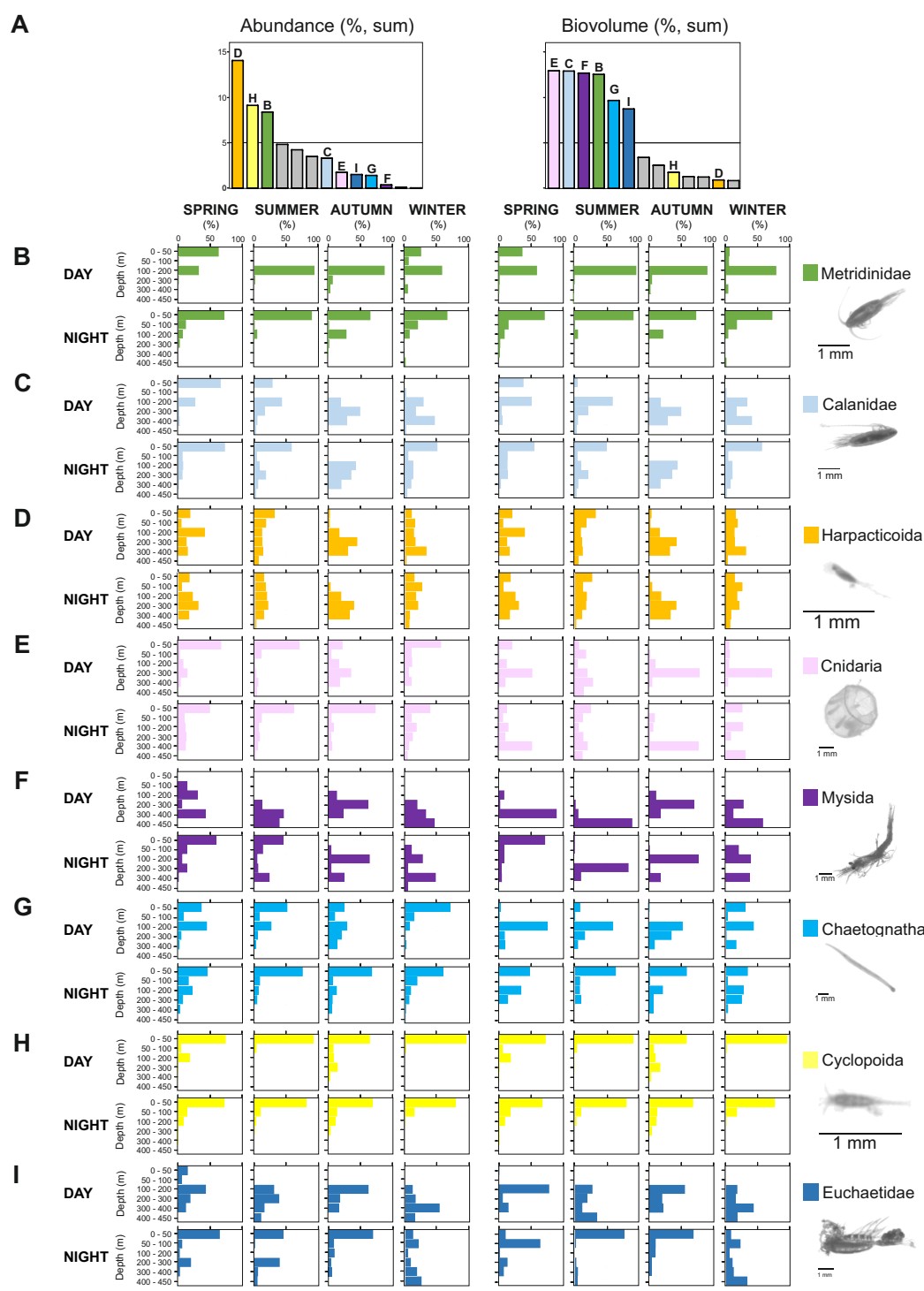

**Figure 7 Seasonal, diel and vertical distribution of the most important taxa.** Vertical, diel and seasonal distribution of the most important taxa representing the percent of abundance and biovolume (%) in relation to their total abundance and biovolume. (A) Total sum of biovolume and abundance in all samples. Vertical distributions of (B) Metridinidae, (C) Calanidae, (D) Harpacticoida, (E) Cnidaria, (F) Mysida, (G) Chaetognatha, (H) Cyclopoida and (I) Euchaetidae. The group "calanoids (<1.5 mm)", although more than 5% in both, biovolume and abundance, is not represented here as it is composed of a mix of taxa with different functions. Pictures scale bar = 1 mm.

Highest abundances of copepod nauplii and cnidarians were found in early spring with a minimum in autumn and raising up again in late-winter. Mysida presented a stable abundance throughout the year with a minimum in summer. These groups clearly presented different vertical distributions (Fig. 7). Metridinidae, a family of large copepods with high biovolume and abundance, resided generally above 200 m, exhibiting a peak between 100-200 m during the day. At night, 74% of their abundance and 77% of their biovolume were found in the upper 50 m (Fig. 7B). On average, larger individuals ($0.62 \pm 0.3$ mm$^3$; average size $\pm$ SD) were found in 100-200 m water depth during day and night. In spring, a significant proportion of the Metridinidae population was found during the day in shallow waters where smaller individuals ($0.14 \pm 0.08$ mm$^3$) were observed. At nighttime, a fraction of those large and small individuals residing at intermediate waters were found also in shallow waters (Fig. 8A). The Calanidae were found mainly in intermediate waters (100-300 m) during daytime, except in spring, where on average larger individuals ($1.37 \pm 0.5$ mm$^3$) resided (Fig. 8B). In spring, small specimens of Calanidae ($0.30 \pm 0.003$ mm$^3$) were found in shallow waters also during daytime (Fig. 8B). At nighttime, half of the population (61% and 54% of their abundance and biovolume, respectively) were present in shallow waters. In autumn, however, the whole population stayed between 100–400 m. Despite the low number of specimens of the Calanidae found at shallow waters, they contributed considerably to the biovolume in these layers (Fig. 6). Euchaetidae showed higher abundance and biovolume in the deep part of the water column during day time (100–450 m), ascending to shallower waters at night (Fig. 7I). Small copepods of the taxa Harpacticoida and Cyclopoida were very abundant, but, as expected, they only represented a small fraction of the total biovolume. Harpacticoida were distributed through the whole water column (Fig. 7D). Cyclopoida were mainly present in the upper 50 m during day and night in all seasons (Fig. 7H). Cnidarians were overall not very abundant, but constituted an important fraction of the biovolume, especially in 200–300 m depth during day in all seasons, and 50–100 m depth during day time in summer and autumn. Their highest abundances were generally found in 0-50 m during day and night, while higher biovolumes were detected in deeper layers 200–450 m (Fig. 7E). Throughout the year, Mysida were mainly present in >200 m depth during the day, accounting for 52–100% of its abundance and 89–100% of its biovolume. In spring, at night a high proportion of their abundance and biovolume were found in shallow waters while in summer only a fraction of their abundance was found in 0-50 m. In autumn and winter the shallowest depth mysids were found was up to 50 m both during day and night (Fig. 7F). Chaetognatha presented the highest abundances in the 0–50 m water depth during day and night in all seasons. Their biovolume, however, peaked at 100–300 m during day and at 0–50 m during night (Fig. 7G).

Overall, the four predictor variables (fluorescence (Chl *a*), oxygen, temperature, salinity) explained 44.3% and 33.69% of the total variation in abundance and biovolume, respectively (Fig. S1). The first axis of the RDA explained 34.15% (abundance) and 25.4% (biovolume) of the total variation while the second axis only accounted for 10.15% (abundance) and 8.29% (biovolume). Relationships between the parameters and the first RDA axis were highest for Chl *a*, while temperature was related to the second RDA

**Table 2** Integrated abundances (ind m$^{-2}$) of the most important zooplankton groups over the entire water column sampled on four dates throughout a year.

| | | SPRING | | SUMMER | | AUTUMN | | WINTER | |
|---|---|---|---|---|---|---|---|---|---|
| | | Day | Night | Day | Night | Day | Night | Day | Night |
| Cyclopoida | | 47,894 | 28,198 | 5,974 | 6,501 | 2,227 | 4,342 | 3,209 | 3,477 |
| | Mean | 38,046 | | 6,237 | | 3,284 | | 3,343 | |
| Harpacticoida | | 48,632 | 47,978 | 9,664 | 12,421 | 9,929 | 7,486 | 6,514 | 4,345 |
| | Mean | 48,305 | | 11,042 | | 8,707 | | 5,429 | |
| Copepoda (nauplii) | | 20,600 | 18,129 | 1,481 | 2,583 | 431 | 2,094 | 2,463 | 1,616 |
| | Mean | 19,365 | | 2,032 | | 1,263 | | 2,039 | |
| Metridinidae | | 27,772 | 9,287 | 25,691 | 8,105 | 12,696 | 18,262 | 4,438 | 1,684 |
| | Mean | 18,529 | | 16,897 | | 15,479 | | 3,061 | |
| Calanidae | | 11,288 | 7,447 | 5,898 | 2,827 | 3,622 | 3,087 | 1,977 | 889 |
| | Mean | 9,367 | | 4,363 | | 3,354 | | 1,433 | |
| Euchaetidae | | 1,840 | 4,259 | 1,731 | 1,110 | 790 | 873 | 1,060 | 733 |
| | Mean | 3,045 | | 1,420 | | 831 | | 896 | |
| Cnidaria | | 4,895 | 2,546 | 1,520 | 3,713 | 702 | 2,432 | 4,636 | 2,848 |
| | Mean | 3,721 | | 2,616 | | 1,567 | | 3,742 | |
| Chaetognatha | | 3,108 | 1,216 | 2,528 | 1,187 | 884 | 1,390 | 1,000 | 598 |
| | Mean | 2,162 | | 1,858 | | 1,117 | | 799 | |
| Mysida | | 239 | 171 | 62 | 164 | 151 | 411 | 239 | 166 |
| | Mean | 205 | | 113 | | 280 | | 202 | |

axis. For both abundance and biovolume the main differences were observed between day and night samples in shallow waters (0–50 m), during spring and summer. The abundance and biovolume of the major zooplankton groups were positively influenced by Chl *a* concentration, except for mysids and ostracods, while temperature did not seem to have a strong influence on the majority of the groups, except for the large zooplankton (amphipods, Euphausiacea and euphausiids) (Fig. S1).

# DISCUSSION

## Physicochemical properties and mesozooplankton seasonal dynamics

The physicochemical observations in our study are consistent with earlier descriptions from Comau Fjord, a temperate Fjord connected to the Pacific Ocean by the Chacao Channel and Ancud Gulf. Comau Fjord has an estuarine circulation and is characterized by a strong pycnocline, where surface waters (0-50 m) present higher variability than the deeper and quasi-homogeneous water layer (Fig. 2). Clear differences were observed in temperature, pH, oxygen and Chl *a* in relation to the season, likely caused by the strong seasonal variability in solar radiation (maximum between spring and summer), precipitation and river discharge (maximum in late autumn and winter) (*González et al., 2010*).

Pronounced seasonality of environmental variables often results in high biological production and are associated to seasonal changes in the holoplankton community (*Mauchline, 1998*; *Balbontín & Bustos, 2005*; *Aracena et al., 2011*), and meroplankton

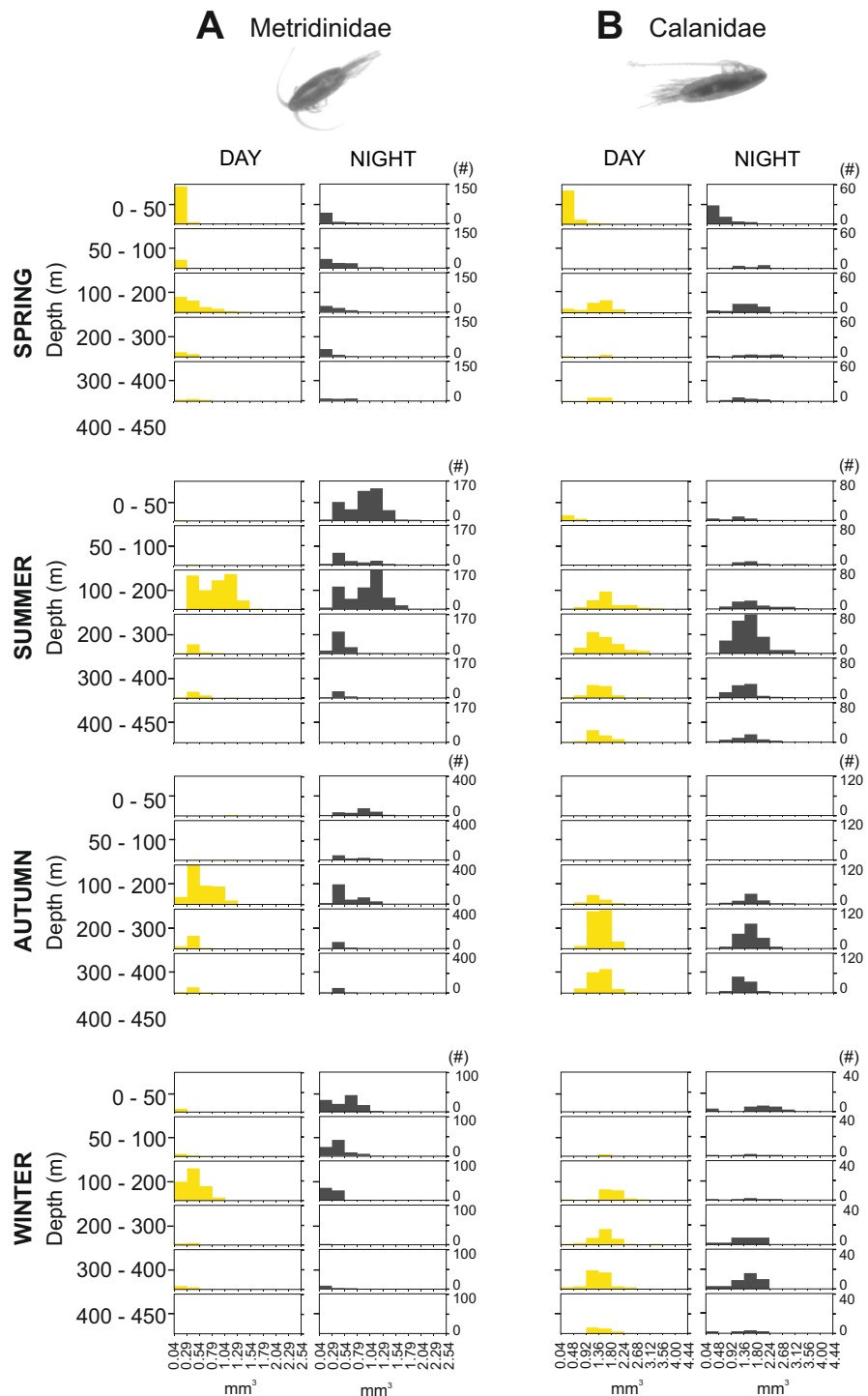

**Figure 8  Size frequency distribution of Metridinidae and Calanidae (Copepoda).** Seasonal, diel and vertical size-frequency distribution of (A) Metridinidae and (B) Calanidae. (#) provides the number of individuals measured.

abundance (*Ladah et al., 2005*; *Landaeta & Castro, 2006*). In Comau Fjord, a thermal inversion of the surface water (0–50 m) in winter is visible (Fig. 2), probably due to heat loss in the surface layer caused by winds and the discharge of cold freshwater from rivers and glaciers (*Silva, Calvete & Sievers, 1997*). Later in the year, the thermal density stratification stabilizes the water column, triggering spring phytoplankton blooms (*Iriarte et al., 2007*), which usually follow rain events and thus the input of nutrients. This leads to a strong increase in Chl *a* (Fig. 2E). As shown by previous studies in the area (*Palma & Silva, 2004*; *Vargas et al., 2008*; *González et al., 2010*), this peak in Chl *a* was most likely due to blooming chain-forming diatoms. The high phytoplankton biomass is expected to be grazed predominantly by copepods, increasing their biomass and establishing the classical diatom-to-zooplankton food web (*Palma & Silva, 2004*; *Vargas et al., 2008*; *González et al., 2010*). Accordingly, the population dynamics of copepods in this study followed the phytoplankton seasonal cycle, with the highest abundance in spring, associated with the maximum concentration in phytoplankton (Table 2). At this time, copepod nauplii and young stages of calanoid copepods, which are indicative of intense zooplankton reproduction, accounted for a large proportion of the zooplankton community. Considering that copepods are the main food source for cnidarians and chaetognaths (*Palma & Kaiser, 1993*), the higher abundance of carnivorous zooplankton occurring in spring (Table 2) can be attributed to the large copepod abundance at this time of the year. In summer, biovolume and biomass reached their maxima (Fig. 3), which together with a lower abundance, suggests the presence of larger individuals or different groups including species with larger individuals. Subsequently, in autumn and winter, zooplankton abundance, biovolume and biomass decreased. At this time, primary production should be low, as reflected by low Chl *a* values (Fig. 2E), and likely zooplankton growth was limited by food availability (*Escribano et al., 2007*). The overall decline in copepod abundance from spring to winter may be explained by the decrease in phytoplankton occurrence and the increasing predation pressure exerted by carnivorous zooplankton. In winter, the plankton in the Fjord typically shifts towards a microbial loop based community grazed by heterotrophic nanoflagellates, which become the main mesozooplankton prey (*Vargas et al., 2008*; *González et al., 2010*) however, this does not support a high secondary production.

The northern part of the Chilean fjord region, the area between Puerto Montt and Guafo's Mouth, represents the most productive area of Patagonia in terms of primary production and zooplankton biomass (*Palma, 2008*). In contrast, the phytoplankton production in the southern area is low due to the stronger influence of glaciers, resulting in cold, fresh and turbid waters (*Palma & Rosales, 1997*; *Palma & Silva, 2004*; *Iriarte et al., 2007*; *Palma, 2008*), and consequently low zooplankton survival and growth (*Giesecke et al., 2019*). Previous studies described ranges for zooplankton biovolume, expressed as the plankton wet volume, of 65 to more than 1,386 ml zooplankton 1,000 m$^{-3}$ outside Comau Fjord, in the Inner Sea off Chiloé island (*Palma, 2008*). This is in line with the present results (250–1,500 ml zooplankton 1,000 m$^{-3}$), showing an especially high biovolume during summer. *Palma & Rosales (1997)* also found the highest values of zooplankton biovolume in the northern part (interior of Reloncaví Fjord and Ancud Gulf) with values that ranged between 56 and 1,626 ml zooplankton 1,000 m$^{-3}$, but a low zooplankton biovolume in the

inner of Comau Fjord. The observed variations may potentially be due to (a) interannual differences with a much lower Chl *a* concentration during the same season of their year of study (*Ramírez, 1995*); and/or (b) methodological and analytical differences, *e.g.,* different sampling gears and proxies for biomass estimation (*i.e.,* measurement of zooplankton wet volume *vs.* image analysis in this study). Moreover, the vertical hauls used in this study may most likely underestimate the abundance of very motile organisms able to avoid nets, such as adult euphausiids, megalopae of *Munida gregaria* or large fish larvae, which are also important in terms of abundance and biomass across west Patagonia (*Antezana, 1999*).

In Fjord systems, oceanographic processes such as estuarine circulation, tidal mixing or water column stratification may influence the composition and abundance of zooplankton communities (*Palma & Silva, 2004*; *Sánchez, González & Iriarte, 2011*). Overall, copepods were the main contributors to the total biomass and biovolume of the zooplankton community, especially during summer (69–78%). This is in agreement with previous studies showing that in Chilean Fjords, planktonic crustaceans, such as copepods and euphausiids, have the highest abundances and biomasses, followed by chaetognaths and gelatinous plankton (*Defren-Janson, Schnack-Schiel & Richter, 1999*; *Palma & Silva, 2004*). Copepods are the most abundant and diverse component of marine zooplankton worldwide (*Mauchline, 1998*), and the abundance of small copepods (<1.5 mm) generally surpasses the abundance of larger ones (*Fransz, 1988*; *Gallienne & Robins, 1998*; *Gallienne, Robins & Woodd-Walker, 2001*). Similarly, small copepods accounted for 58–86% of the total copepod community in the present study. Another important contributor to zooplankton communities in Chilean Fjords is the euphausiid *Euphausia vallentini*. The present study revealed young stages of euphausiids during spring and summer but only one adult specimen in winter, indicating that euphausiids are present in Comau Fjord, but were not caught efficiently in our samples. This is likely related to the small volume filtered by our net, the patchy distribution of *E. vallentini*, and their ability of avoiding slow nets (*Brinton, 1962*). Like euphausiids, mysids can form dense swarms, making them a potential food resource for a wide range of organisms, from predatory fishes to benthic CWC. Despite their important contribution to the total zooplankton biovolume in deep waters (Fig. 6), poor attention has been given to their presence in Chilean Patagonia. To our knowledge, there are only two studies describing mysids in this area: *Guglielmo & Ianora (1997)* found that the most abundant species for the Strait of Magellan is the deep-dwelling *Boreomysis rostrata*; *Díaz-Astudillo, Cáceres & Landaeta (2017)* found higher abundances of mysids during night and inside the Reloncaví Fjord and Ancud Gulf. Thus, this study constitutes the first record of mysids in Comau Fjord.

## Zooplankton diel vertical distribution and migration

Biological processes (*e.g.,* diel vertical migration, predator avoidance, location of food patches and mating), together with oceanographic processes (*e.g.,* estuarine circulation or water column stratification) are mechanisms by which the underlying zooplankton behavior presents high spatial heterogeneity (*Folt & Burns, 1999*). In the present study, zooplankton abundance, biovolume and biomass significantly differed between day and night. These differences were probably due to the high patchiness and the vertical distribution the

zooplankton exhibited during day and night, especially by larger zooplankton individuals, those able to form swarms (*i.e.,* mysids and euphausiids) and migrate (Figs. 7B–7I). For instance, Euchaetidae, Euphausiacea, Metridinidae and Mysida, four of the larger and most important zooplankton groups, changed their centroid depth between day and night contributing to 30–40% of the day-night differences in abundance and 30–38% in biovolume (Fig. 4). To a lesser extent, small organisms, *e.g.*, Appendicularia, Clausocalanidae and Copepoda (nauplii) also contributed to the these differences, especially in summer and autumn, however, their contribution was lower (0–29%) in comparison to larger zooplankton organisms.

Diel Vertical Migration (DVM) is usually associated with differences in light intensity within the photic zone, taking place periodically in 24 h cycles (*Brierley, 2014*). During daytime, zooplankton organisms migrate to deeper, darker waters to avoid visual predators, such as fishes, while they come to the surface for feeding at night (*Hays, Webb & Frears, 1998*). In Comau Fjord, DVM seemed to be related to the size of the zooplankton. Here, large zooplankton taxa, particularly large copepods, that inhabited the intermediate waters (100-300 m) during daytime ascended to shallow waters at night, while small organisms, mainly cyclopoids and harpacticoids, did not perform a clear DVM (Figs. 7B–7I). Throughout the year, Cyclopoida centroid depth fluctuated between 40-85 m water depth and Harpacticoida between 149-254 m water depth. These two copepod groups, however, differed in their vertical distribution. The highest abundances of cyclopoid copepods were mainly found in the upper 50 m, whereas harpacticoids were found throughout the water column (Fig. 7D).

Large individuals from several calanoid copepod taxa (Metridinidae, Calanidae, Euchaetidae), mysids, chaetognaths and cnidarians did perform DVM over 300 m. This agrees with the findings by *Hays (1995)* that DVM is pronounced in large and pigmented species due to their susceptibility of being perceived by visually orientated predators. Similarly, studies in northern Patagonia have shown that conspicuous zooplankton organisms tend to avoid well illuminated waters (*Villenas, Soto & Palma, 2009*). According to *Hays, Kennedy & Frost (2001)*, large individuals of *Metridia* usually reside in deep waters, and only a fraction of these ascends to shallow waters at night, whereas smaller individuals stay at the surface. In Comau Fjord, Metridinidae showed the highest values of biovolume and abundance at intermediate depths (100–200 m) during daytime, but in the surface layer (0–50 m) at night, indicating that 77% of the entire population migrated towards the surface (Fig. 7B). In spring, a proportion of the Metridinidae population was found in shallow waters during the day. This was due to the smaller size of the individuals (Fig. 8A) and the higher amount of food available here. Following the same pattern as Metridinidae, larger individuals of Calanidae were found in deeper waters during daytime, whereas small specimens were found in shallow waters also during daytime in spring, where they may not be hunted by visual predators due to their small size. It is possible that those small organisms were individuals of earlier life stages (*i.e.,* small) or species with a smaller size. Euchaetidae also performed DVM with most of the organisms living continuously in deep waters (100–450 m) during the day, but a large proportion of the population (60% and

53% of their abundance and biovolume, respectively) migrated to the surface during night, except in winter (Fig. 7I).

Mysids accounted for a big proportion (up to 70%) of the total zooplankton biovolume, especially in spring and winter (Fig. 6). During day, they were generally detected in deep waters (>200 m water depth) and at night they migrated upwards, although their DVM pattern was only clear in spring and summer. At night in spring, both abundance and biovolume were high in 0–50 m water depth, whereas in summer only a high abundance was detected at that water depth. The fact that biovolume peaked at 200–300 m in summer at night suggested that only small individuals migrated to shallow waters while larger ones stayed at deeper waters (Fig. 7F).

Chaetognaths were distributed throughout the water column with the highest biovolumes between 100–300 m, during both day and night (Fig. 7G). This is in accordance with the distribution found by *Guglielmo & Ianora (1995)* for the Strait of Magellan. A particularly high abundance of chaetognaths in the upper 0–50 m during both day and night and a high biovolume only at night indicated that larger individuals migrated to shallow waters at night likely to feed on other organisms. South of Comau Fjord, between Guafo's Mouth and the Pulluche Channel, the vertical distribution of cnidarians (*i.e.,* Hydromedusae and Siphonophorae) showed highest abundances in the upper 100 m, specifically in the 20–50 m stratum (*Palma, Apablaza & Soto, 2007*). In this study, cnidarian highest abundances were found in 0–50 m during day and night. However, biovolume peaked in every season, except for summer, at 200–300 m, indicating that larger individuals of cnidarians stayed at deeper waters while small individuals resided in shallow waters.

In the 50–100 m water layer a minimum in zooplankton abundance and biomass was found in all seasons (Fig. 5). This "zooplankton gap" could be related to a high concentration of predators in this water depth. In northern Chilean Fjords, the high biomasses and abundances of gelatinous organisms are correlated to a decrease in chitinous biomass of other organisms (*Palma & Silva, 2004*; *Villenas, Soto & Palma, 2009*). It is known that chaetognaths and cnidarians can grow at fast rates, forming dense aggregations that seasonally dominate the zooplankton biomass (*Casanova, 1999*; *Brodeur, Sugisaki & Hunt Jr, 2002*) by feeding voraciously on copepods and larvae from other organisms (*Lie et al., 1983*; *Palma & Rosales, 1997*). In our study, biovolume of carnivorous organisms ranged between 18 and 83% of the total zooplankton in this water layer, reaching particularly high values in summer and autumn (50–83%) (Fig. 9). These previous evidences suggest that carnivorous organisms may have reduced zooplankton abundance in this depth stratum and predation may play a role in structuring the vertical zooplankton community in Comau Fjord.

Mesozooplankton plays an important role in the food web as a food source for many pelagic and benthic organisms (*González et al., 2013*), including CWC (*Gili et al., 2006*; *Carlier et al., 2009*; *Mayr et al., 2011*). Due to the difficulty of studying CWC *in situ*, little is known about their natural diet and its availability. In Comau Fjord, CWC thrive also in deeper, naturally acidified waters (*Häussermann & Försterra, 2007*; *Försterra, Häussermann & Laudien, 2017*). Although the environment is unfavorable, this might be due to the high ingestion rate (*e.g.*, of the CWC *Desmophyllum dianthus*) which showed a positive impact

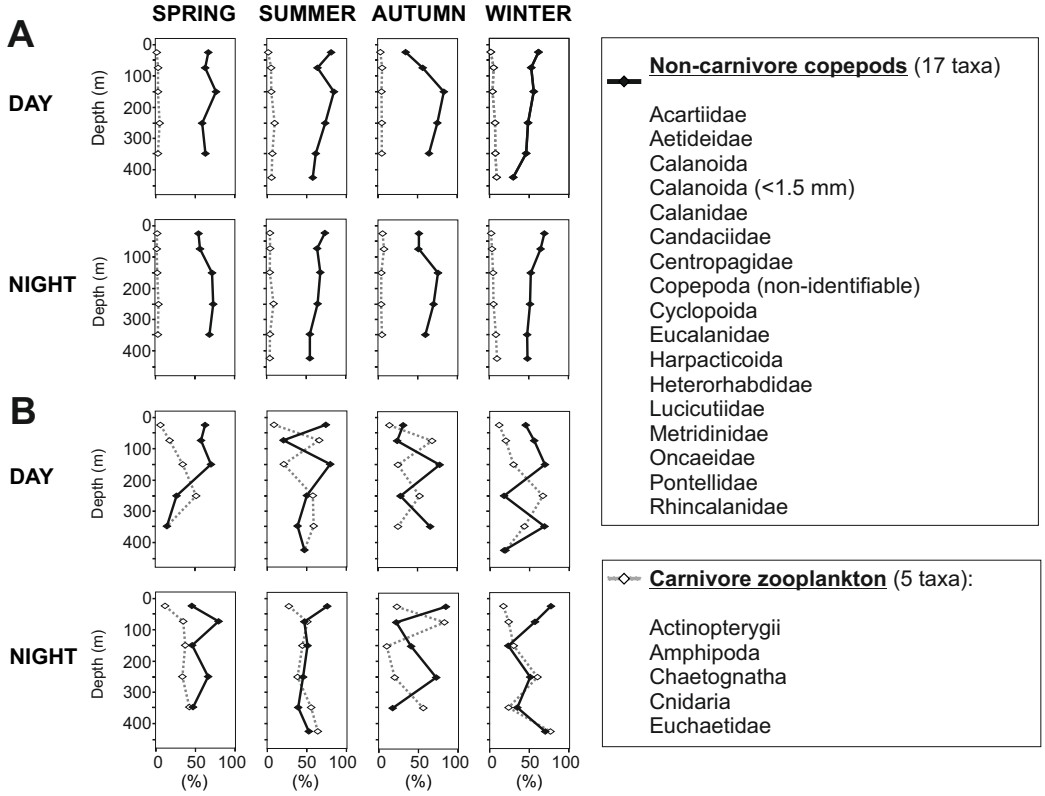

**Figure 9** **Non-carnivore copepods *versus* carnivore zooplankton.** Presented as percentages of (A) total abundance and (B) total biovolume.

on their calcification rates, regardless of the seawater pH (*Martínez-Dios et al., 2020*). Zooplankton abundance and biomass are highly influenced by seasonality. Therefore, CWC must be adapted to differences in food availability, *i.e.,* high zooplankton abundances during spring and summer and low concentrations during autumn and winter. Indeed high growth rates of *D. dianthus* were found in summer (*Jantzen et al., 2013b*) which may be associated with higher zooplankton availability. In winter, when zooplankton biomass is low, CWC may slow down their metabolism to cope with the lower food availability (*Naumann et al., 2011*). Only recently, a study confirmed that *D. dianthus* preyed on medium and large sized calanoid copepods and euphausiids (*Höfer et al., 2018*). Based on the present DVM data, shallow-dwelling CWC in Comau Fjord might feed on small copepods (*e.g.,* cyclopoid and calanoid) during the day, and on larger organisms during night hours when zooplankton migrates upwards. Deeper-dwelling corals, by contrast, may mainly encounter larger prey, such as mysids or large calanoid copepods (Calanidae, Euchaetidae) and thus may gain enough energy to upregulate their internal pH in an acidified environment.

## CONCLUSIONS

The seasonal changes of zooplankton over the entire water column showed that abundance peaked in spring, likely due to spring phytoplankton blooms. In summer, biovolume and biomass were the highest and decreased thereafter over time, reaching the lowest values in late autumn and mid-winter. The low concentration of Chl *a* during the cold season suggests that primary production was insufficient to support high levels of secondary production. The vertical distribution of zooplankton biovolume and biomass differed between day and night, with a daytime maximum between 100 and 200 m water depth and a nighttime maximum in surface waters (0–50 m) associated with the diel vertical migration of calanoid copepods of the family. Overall, copepods were the dominant group of the total zooplankton community with an important contribution of small organisms (individuals <1.5 mm), followed by mysids, chaetognaths and cnidarians (biovolume and biomass), and nauplii and Appendicularia (abundance). The integrated abundance, biovolume and biomass also showed significant differences between daytime and nighttime values. These differences were probably due to the high zooplankton patchiness driven by both biological and oceanographic processes. Diel vertical migration, predation avoidance, location of food patches as well as estuarine circulation, tidal mixing or water column stratification are considered to be the main drivers of the zooplankton distribution in Comau Fjord.

## ACKNOWLEDGEMENTS

We thank the team of PACOC which contributed to the success of the project. Support during sampling was provided by the scientific and logistic staff from the Fundación San Ignacio del Huinay. Henning Schröder and Manding Suwareh assisted with the sample collection. Thomas Heran, Simon Schöbinger and Jasmin Stimpfle helped with the treatment, analysis and scanning of the samples. Gertraud Schmidt-Grieb provided support in an early phase of the project.

### Funding

This study was funded by the bilateral Chilean-German PACOC project (CONICYT-BMBF 20140041; BMBF 01DN15024) as well as CONICYT FONDAP-IDEAL 15150003 and AWI (PACES II, Topic 1, WP6; Changing Earth–Sustaining our Future: Subtopic 6.1). The funders had no role in study design, data collection and analysis, decision to publish, or preparation of the manuscript.

### Grant Disclosures

The following grant information was disclosed by the authors:
The bilateral Chilean-German PACOC project: CONICYT-BMBF 20140041, BMBF 01DN15024.
CONICYT FONDAP-IDEAL 15150003.

AWI (PACES II, Topic 1, WP6; Changing Earth–Sustaining our Future: Subtopic 6.1).

## Competing Interests
The authors declare there are no competing interests.

## Author Contributions

- Nur Garcia-Herrera conceived and designed the experiments, performed the experiments, analyzed the data, prepared figures and/or tables, authored or reviewed drafts of the paper, and approved the final draft.
- Astrid Cornils and Barbara Niehoff performed the experiments, analyzed the data, authored or reviewed drafts of the paper, and approved the final draft.
- Jürgen Laudien and Juan Höfer performed the experiments, authored or reviewed drafts of the paper, and approved the final draft.
- Günter Försterra and Humberto E. González conceived and designed the experiments, authored or reviewed drafts of the paper, and approved the final draft.
- Claudio Richter conceived and designed the experiments, analyzed the data, authored or reviewed drafts of the paper, and approved the final draft.

## Data Availability

Data is available at PANGAEA, accession link:

https://www.pangaea.de/tok/6c057d3563805b430c52fffcce8eda6b429ed702.

## Supplemental Information

Supplemental information for this article can be found online at http://dx.doi.org/10.7717/peerj.12823#supplemental-information.

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
