# Peer review of "Seasonal and diel variations in the vertical distribution, composition, abundance and biomass of zooplankton in a deep Chilean Patagonian Fjord"

_PeerJ, doi:10.7717/peerj.12823_

## Round 0.1 · original submission · Major Revisions

I have got two very good constructive reviews of the paper. Both praise the work while making considerable suggestions to improve the paper. Please consider their comments thoroughly. In particular, do not oversell the paper (PeerJ is interested in good science, not inflated titles or statements like some journals that seek media attention). For example, if corals were not studied then they may merit a mention as useful context but not in the conclusions. We look forward to receiving your revised manuscript in a few weeks.

·

Basic reporting

The submitted manuscript reports the vertical distribution of (meso)zooplankton at a fixed station in a deep fjord of northern Patagonia (Chile). Zooplankton was sampled at four different dates covering all four seasons and samples were analysed with imaging approaches (ZooScan). Environmental data were also collected to help interpret the zooplankton distribution. The limited sampling (four dates in one year), does not allow to infer a true seasonal cycle (which would have required replication of the season sampling), but the information brought by the collected data is interesting allowing to focus on potential diel vertical migration, which is at the core of the study.

The rationale of the study is presented in the much broader context of the sustainability of cold water corals (CWC) which face calcification problems due to aragonite-undersaturated local waters. Their growth and survival depend on the availability of food resources that mainly consist in zooplankton, hence the question of its composition and distribution. If I acknowledge this broad context, I however consider that the study did not investigate the relationship between the diversity and distribution of the zooplankton and the sustainability of CWC populations, which is presented as the second objective of the study. The CWC are not studied in the paper, and only speculative hypotheses are inferred in the discussion section (no materials and methods, no results on CWC). Further comments on this point will be given in the following sections.

Overall the manuscript is written in an English that a non-native speaker can understand. Some ambiguous terms are reported in the detailed comments.

Some sections are a bit long and might be shortened. In the results section for example, line252-265, the paragraph is too detailed, and information is found in Fig. 4. I suggest to keep only key results. Overall the discussion is long and should be shortened. Many details are given, sometimes redundant with the results, with no real discussion. Shortening could be achieved by refocusing some paragraphs (e.g. 2nd paragraph should focus on vertical distribution and DVM, nothing else), and by removing parts which seem out of topic. As an example of this last point, part of the mysid paragraph (in second subsection of the discussion) (lines 458-464) could probably be removed, as this do not relate to vertical migration. So please try to keep only essential discussion. Similarly, the “zooplankton gap” part (lines 478-500) is very interesting but is too detailed given the results available, and should be shortened.

The Figures and Tables provided are all helpful and are overall well presented. Some detailed comments are provided in the following sections.

Raw data were accessible through the Pangaea repository. Data available consist of images of each ZooScan scan. Other data, like a table of counts of each plankton category, i.e. after image processing, or the accompanying CTD data, are not available. I don’t know whether this is mandatory.

Experimental design

The sampling design allowed to answer the question of the vertical distribution of various groups of zooplankton in the deep fjord, and to search for the existence of diel vertical migration patterns. One question was however why no flowmeter was used to measure more accurately the volume filtered. Is it not possible when sampling multiple strata?

As mentioned in the basic reporting, the study did not investigate directly the relationship between the zooplankton and the coral populations.

Some details in the description of the image processing were lacking or some points were not clear. These are higlighted in the attached manuscript. Among them are:
1. “ZooScan detection limit was set at the standard of 300 µm” (line 161-162): what does this mean?
2. You mentioned several times (lines 163, 179), the success of automatic processing or classification. What does this mean, and how is it determined?
3. Was the training set you used a combination of the global one and the Patagonian one, or only the Patagonian one?

Statistics. No statistical analyses were provided in this paper. In particular, in the analysis of the vertical distribution of zooplankton (with abundance), it is not clear if the seasonal or diel differences were significant or not. One example is the case of Calanus / Calanidae (lines 306-308, fig. 6C), in which day-night difference is not so clear; this is found in many other examples… Although I can agree that day-night differences are visible in Fig. 4 for biovolume and biomass, it is not so evident for abundance, so that statistical tests might be of help. The lack of statistics could arise from the lack of replication (which is not a problem per se for me because of the sampling constraints…), so the test of Beet et al. (2003) could not be applied, but maybe the method proposed by Solow et al. (2000) could be a solution. I am not sure if such an approach could also be used for biovolume and biomass, but certainly it could be used for abundance. Could you explain why you didn’t/couldn’t use any statistical test?
Beet, A., Solow, A. R. & Bollens, S. M. (2003) Comparing vertical plankton profiles with replication. Mar. Ecol. Prog. Ser. 262:285-287.
Solow, A. R., Bollens, S. M. & Beet, A. (2000) Comparing two vertical plankton distributions. Limnol. Oceanogr. 45(2):506-509.

Validity of the findings

1. As already mentioned, the last paragraph of the discussion, "Ecological connection with cold-water corals” is very speculative and should be greatly shortened, if not removed. You did not study the relationship between CWC and mesozooplankton. Yes there is plankton that could serve as a food source for CWC. Yes zooplankton is more abundant in spring which is also the period of higher growth rate for CWC. But your data you don’t allow to show the relationship between both. Further you cannot show that DVM is important for CWC feeding: for this you would have needed feeding studies comparing shallow and deep corals, and day and night food consumption. Finally, I did not understand the calculations of prey consumption that allowed you to provide a threshold for prey availability that would be necessary to sustain a coral population. Again this sounds highly speculative. Similarly, the second part of the conclusion section is out of the core topic: the manuscript did not show that “The high amount of zooplankton in Comau Fjord provides sufficient nourishment to maintain a viable coral population despite the aragonite under-saturated waters”. This remains a hypothesis.

2. As mentioned in the previous section, if statistical analysis could be done, this would give more strength to the conclusion about DVM.

3. Line 300 and following lines: in this part of the paragraph you describe the vertical distribution of different size groups within one taxon (e.g. “largest Metridia”, “small specimens of Calanus”), but such results are not illustrated in any figure or table. This is however a very interesting result of your study. In the discussion you conclude (lines 437-438) that “It is possible that the size differences between day and night might be a consequence of the presence of different species or life stages”. This deserves more discussion, based on size data. Why not show size data? I am not talking here of the total biovolume of a taxon, but the vertical distribution of different size classes within this taxon, so that you can show their potential different distribution.

4. Taxonomy-1. I am not comfortable with the term “taxonomic” throughout your manuscript because the defined groups mix true taxonomic groups (e.g. Copepoda, Cirripedia) and developmental stage groups which can mix several taxonomic groups (e.g. crustacean nauplii which can include copepod nauplii, as you mentioned line 272, and cirriped nauplii; what then is included in the group Cirripedia? Cyprids?). I have no ideal solution, but you can either avoid the terms “taxonomic” or precise somewhere that what you call taxonomic is not really taxonomic… Please note that this may have consequences in the interpretation of the results in the discussion. For example when you state that “At this time, crustacean nauplii and young stages of calanoid copepods, which are indicative of intense zooplankton reproduction” (lines 355-356), this is only partly true, as crustacean nauplii can also indicate reproduction of benthic cirripeds as you (too) briefly recognize line 368 when mentioning meroplankton. Such a point would merit more discussion, as abundances of barnacle larvae may reach values similar to yours: for example, Meerhoff et al. 2014 reported abundances of barnacle larvae of up to 80 larv m-3 in october, of course in a different fjord.
Meerhoff E, Tapia FJ, Castro LR (2014) Spatial structure of the meroplankton community along a Patagonian fjord - The role of changing freshwater inputs. Prog Oceanogr 129:125-135.

5. Taxonomy-2. It is sometimes difficult to follow your statements on the taxonomic composition as taxonomic levels are different depending on the groups and depending on your statements. For Copepoda, sometimes you use the order (e.g. Cyclopida), sometimes the family (e.g. Euchaetidae). More importantly there are some differences between the text and the figures: for example when lines 290-294 you mentioned Metridia (also in Table 2), citing figures 5 and 6, the corresponding figures did not consider Metridia but Metridinidae. The same applies to lines 300-306. This could mean that Metridia is the only genus observed in this family or the most abundant one, but this brings confusion. Could you please make some efforts to keep things more homogeneous? The same comment can be made for Calanus, lines 306-312, while fig. 6C illustrates the family Calanidae. Please also update in the discussion accordingly (lines 422-436).

Additional comments

1. Table 2: you cannot sum day and night abundances, because it artificially increases the abundance: the organisms present at nighttime are not added to those of the daytime, they replace them. So I suggest using a mean instead.

2. I suggest you avoid beginning your discussion by sentences like “The present study provides the first detailed data on the zooplankton community composition, distribution and dynamics in the northern Chilean Patagonia. There are only few zooplankton records with a high taxonomic resolution in Patagonian Chilean waters.” The interest of your study does not rely only on the fact that this is the first in your area… I also find not very convincing the argument on high taxonomic resolution as your own study does not provide high taxonomic resolution: at best you identified genera but all your figures and tables address much higher levels…

3. Lines 357-358: “During summer, biovolume and biomass reached their maxima (Fig. 3), indicating the growth of the zooplankton organisms.” I think that you cannot reliably draw this conclusion. First this has to be put in relation with the lower abundance in summer, and not only by itself (lower abundance + higher biomass, then larger individuals). Second, increase in biovolume/biomass not necessarily reflect growth, but may reflect different groups, with larger species, so this is not growth. If this is this idea that you had in mind, then I think that the word growth is inadequate.

4. There is a mix between the 2 first paragraphs of the discussion section, which could bring some confusion. For example lines 440-445 would be better in the first paragraph “mesozooplankton seasonal dynamics” than in the second one although it is of course “population dynamics”. The second paragraph should be focused on vertical distribution/migration. Lines 440-445 have nothing to do with vertical distribution. Same for lines 472-477…

5. Lines 501-510: this paragraph is very important, as looking at results described lines 247-251 immediately raises the question of the factors responsible of this. However, biological factors are probably not the only ones to explain this observation, and other factors like tides or others (those you mentioned lines 386-388) may be involved to transport plankton to the sampling site (in relation to the patchiness discussed in this paragraph). You may say something here.

6. Line 492/Fig. 7: You give a range of 68-82% for carnivores. But even in summer and autumn, it can be less (summer night). For the other months, it is much less, so you cannot simply give the range that you prefer (disproportionately large… in 2 months out 4!). I suggest changing the sentence to “In our study, biovolume of carnivorous organisms ranged between X and Y % of the total zooplankton in this water layer, reaching particularly high values in summer and autumn (Z-82%) (Fig. 7)”. Further, in this figure, it is not possible to contrast copepods and carnivorous zooplankton, because the category carnivorous includes the Euchaetidae, which are copepods (are there other carnivorous copepods?). So I suggest changing the name of the category copepods (e.g. non carnivorous copepods, stating somewhere that Euchaetidae are carnivorous).

7. Detailed comments
Fig. 4: Please add “Abundance (ind m-3)” on the X axis, not as a title (same for biovolume and biomass)
Fig. 6: For all taxa, do the counts include all developmental stages?
Other detailed comments are given in the attached manuscript.

·

Basic reporting

This manuscript deals with the spatio-temporal variability of the zooplankton in a fjord of northern west Patagonia. Using vertical hauls at different depths, with day and night collections during different seasons, and with novel methodologies for the identification and measurement of zooplankters, authors describe the vertical distributions (not the DVM, which is not statistically tested) of several taxa, as well as the CTD casts. Based on this large amount of data, the authors made a very nice description of the data, but it lacks statistical comparisons to validate the observed patterns. Also, no working hypothesis was tested by the authors. The incorporation of several methodologies in the data analysis will improve the manuscript quality. Some suggestions are included in the following sections.
Finally, the relationship between CWC and zooplankton biomass is not tested in the manuscript, and therefore, this should not be included in the Conclusions.

Experimental design

The experimental design of the study is correct, with three factors (depth, day/night, season), but without replicates (no additional hauls for each stratum).
The lack of replicates does not allow the use of two-way ANOVA for testing DVM, but a 2x4 table of contingency (day/night vs season) of the centroid depth may be an alternative for testing DVM.

Validity of the findings

This is the weakest section of the manuscript. The description of the results is sound, but the findings were not statistically tested. I suggest several methods throughout the Results

Additional comments

Introduction
line 119 and its relationship with zoo- and ichthyoplankton growth and feeding (Landaeta et al. 2015a, b)

Landaeta MF, Contreras JE, Bustos CA, Pérez-Matus A (2015a) Growth and condition of larval rockfish in a Patagonian fjord-type inlet: role of hydrographic conditions and food availability. Aquatic Ecology 49: 573-584. doi: 10.1007/s10452-015-9547-y

Landaeta MF, Bustos CA, Contreras JE, Salas-Berríos F, Palacios-Fuentes P, Alvarado-Niño M, Letelier J, Balbontín F (2015b) Larval fish ecology, growth and mortality from two basins with contrasting environmental conditions of an inner sea of northern Patagonia, Chile. Marine Environmental Research 106: 19-29. doi: 10.1016/j.marenvres.2015.03.003

line 227 How do you compare the abundances, estimated biomasses/volume and composition among seasons and depths?
Any correlation/covariance between the biological matrix and the environmental conditions measured by the CTD?

line 230 Any statistical comparison of the surface waters among seasons?

line 235 "mean average". Mean, median and mode are all averages, so "mean" by itself is enough
line 253 Any statiscal comparison?
line 254 Because of the lack of replicates during the diel cycle, authors cannot compare the abundances in the vertical scale using the classical approach (two-way ANOVA with day/night and depth as factors, being the interaction [day/night * depth] the evidence of vertical migration. Otherwise, you can estimate the centroid depths during day/night and seasons, and compare these values in a contingency table.
line 268 Additionally, you can compare the composition and abundance/biomass of zooplankton using two-way PERMANOVA or ANOSIM, and detect those taxa that support dissimilarity using SIMPER
line 273 Also, the multivariate approach to show the depth and seasonal variability of zooplankton composition is a nice option: PCA, PCoA and nMDS are the most common approaches.
line 328 I suggest comparing the biological matrix with environmental conditions measured by the CTD, in order to detect physico-chemical-biological interactions. You can use an univariate approach (multiple regression of environmental conditios against diversity indices), multivariate (PCoA, CCA, PLS) or modelling (GLM, GAM, GLMM)
line 331 Please, indicate the limits of this research (i.e., vertical hauls may reduce the presence of very motile zooplankters, such as adult euphausiids, Munida megalopae, large fish larvae- which are also important in terms of density and biomass across west Patagonia)
line 370 "Balbontín" instead of "Balbotin"
line 388 In terms of tidal mixing, in which lunar condition was carried out the sampling? because there is a noticeable influence of the tidal regime (neap tides vs spring tides) in northern Patagonia, this issue may impact the results and its interpretation
line 416 This "pattern" should be statistically tested
line 485 "this study" I cannot find the correlation test in the manuscript
Finally, the relationship between CWC and zooplankton biomass is not tested in the manuscript, and therefore, this should not be included in the Conclusions

---

## Round 0.2 · Minor Revisions

Thank you for the revisions. Referee 1 makes some additional suggestions which I think are worth addressing (especially re Fig 7 perhaps should be % of species abundance and not total). However, I am not sure if their suggestion regarding fig 5 labels is warranted.

In addition, the figures are a bit cluttered, and their text could be reduced in quality and made larger in places to ensure it is legible:
Figures should have 0, 100, 200 on the vertical axis. Note it is incorrect or misleading to change the scale (and then join the dots in the plot in fig 2 and 9) as this affects the apparent trend (i.e. cannot go 0, 50, 100, and then 100, 200, 300; same with 400-450 as the last band). Also, in fig 9, the carnivore and non-carnivore lines on graphs should be more distinct, such as by using solid/dotted lines and solid/hollow symbols. In Fig 4, avoid using similar colours and use white, hatched, stippled to improve distinctness. Fig 2 horizontal axis needs a much larger (treble) text size.
Add tick marks to axes of figures with numerical scales.

Another issue is the abstract sets out to determine if the presence of corals below the ASH is due to food supply. But the paper never studies corals or their diet nor mentions them in the Conclusions, and the Discussion mentions the corals but has no advance on their diet (the fact that zooplankton are present is not especially novel). That is fine, but please reframe the abstract accordingly. Also, on line 536 the species name should be in italics.

·

Basic reporting

I here acknowledge the thorough revision of the manuscript. All questions/comments I made (as well as those made by the other reviewer) were addressed, most of them satisfactorily (please see in the following further comments). For this I would like to thank the authors.

In particular, they have made required additional analyses to give statistical reliability of their data. I particularly appreciated the RDA analysis, which I think has been well interpreted. Without being completely sure, I think the test of the difference of centroid depth between day and night is appropriate, but still have doubt on how the SIMPER procedure could be interpreted: above which contribution threshold can you consider that the centroid depth differs for a given taxon (then to show significant difference in vertical distribution for this taxon)?

I also aknowledge the huge efforts made to reduce the length of the discussion (almost half has been removed), which allowed to better focus on the core topic of the paper (seasonal variations in zooplankton composition and abundance, and vertical distribution-DVM).

The part dealing with CWC has also been reduced. I would have expected even more reduction, but in the current form this is fine for me

Dealing with data availability, the authors provided additional tables, as was suggested
Discussion - section DVM. I think that before discussing details about DVM, you should first explain that you observe DVM, based on your results. For example based on the RDA, that clearly shows that differences in the zooplankton composition are higher in the surface waters between day and night, which could be due to DVM. Please also use your results from Fig. 7 and if needed the centroid depth and contribution (but see comments above and below). E.g. lines 467-470, it is not clear on what results your statement is based: from your SIMPER analysis, in autumn, calanoid < 1.5 mm contribute to the difference, and the same for copepod nauplii in summer and autumn (fig. 4). Did I misunderstand fig. 4? So it is not only a question of size. I suggest here that you don’t use the general phrase “while small organisms (individuals <1.5 mm)” which implicitly comprises the calanoid < 1.5 mm. Instead, use directly the names cyclopoid and hapacticoid, to state that some copepods do not migrate. Then you can talk about their size…

This might be unfair but I have identified an issue that I did not notice in my first review: in Fig. 7B-I, isn’t it a problem to use % of total zooplankton? If the species of interest does not migrate but that other species migrate downwards, then mathematically its % abundance (or biovolume) will increase at the surface, and decrease at depth. Why not use abundance, or percent of abundance of the species at a given depth in relation to its total abundance (all depths)?

Experimental design

All issues were correctly addressed, in particular for the statistical part (see basic reporting)

Centroid depth: if I understood well you calculated it based on abundance. But from the results you assessed the dissimilarity with abundance and biovolume. Is it possible? In this case why not on biomass? These choices are not clear to me.

Validity of the findings

L. 482-485: “In Comau Fjord, Metridinidae showed the highest values of biovolume and abundance at intermediate depths (100-200 m) during daytime, but in the surface layer (0-50 m) at night (Fig. 7B), which suggested the migration of the largest individuals towards the surface (Fig 8)”. I agree with the first part of the sentence (migration upwards during nighttime) but not sure for the second part: looking at Fig. 8A, summer and autumn for example, all sizes, not only largest, migrate upwards. Am I wrong? (see also additional comments).

Additional comments

L. 282-285 deal with vertical distribution (centroid depth), then should be in the following paragraph…Right?

L. 334-336: not clear. What do you mean by “the largest individuals” with the mean of 0.62… In summer you have individuals larger than this… Further I am not convinced that the size structure is really different between day and night. They seem to be very similar. The main difference in summer is that at night Metridinae can be found in shallower waters, but the “migrating” individuals (i.e. those at 0-50 m) seem to have the same size as those from 100-200 m (not the largest). Do you mean that individuals at intermediate depths are larger in average, i.e. only small specimens are at the surface, and small AND larger are at intermediate depth (and not the largest…)? I am not sure this is what you stated here. So did I misunderstand fig. 8? From this figure I can’t see a major difference in size structure between day and night, for both groups.

L. 344-345: “Euchaetidae showed higher abundance and biovolume in the deeper part of the water column during day time, ascending to shallower waters at night (Fig. 7I)” I can agree looking at the figure, but it is not so obvious… In particular, it seems not true in winter. But, Ok…

L. 542-543: here it is not question of abundance integrated over the water column, it is vertical distribution, as it deals with centroid depth. So please avoid using the term integrated. Moreover, I think that the whole paragraph (542-558) should be place at the beginning of the DVM section, to state that there is vertical migration (change in centroid depth between day and night). This cannot be put after the part on CWC.

Fig. 5: the legends of X-axes (abundance, biovolume, biomass) should be put above the graphs, not below

Table 1. Bivalvia: please add larvae. Same for Gastropoda

RDA: I couldn’t find the legend of fig. S1 but I guess the arrows were the 13 taxonomic group which were the most abundant? Its quality (resolution) should be improved as the different symbols are not easy to distinguish. Instead of “The physicochemical variables have higher influence on abundance and biovolume between daytime and nighttime and within the most productive seasons (spring and summer) in shallow waters (0-50 m)” (L; 362-364), I suggest “For both abundance and biovolume the main differences were observed between day and night samples in shallow waters (0-50 m), during spring and summer”.

·

Basic reporting

The new version of the manuscript has considered all the comments and suggestions made by me in the previous review, and I acknowledge that. This situation has improved all the aspects of the manuscript.

Experimental design

no extra comments

Validity of the findings

Now, with the inclusion of the RDA and centroid depth distribution, the validity of the results has been highlighted

---

## Round 0.3 · accepted · Accept

Thank you for the final revisions and improvements to the figures. Happy new year.